# Experimental Study on a Multi-Evaporator Refrigeration System Equipped with EEV-Based Ejector

**DOI:** 10.3390/e24091302

**Published:** 2022-09-14

**Authors:** Jia Yan, Chen Wang

**Affiliations:** 1School of Civil Engineering and Architecture, Southwest University of Science and Technology, Mianyang 621010, China; 2School of Control Science and Engineering, Shandong University, Jinan 250061, China; 3Department of Mechanical, Aerospace and Civil Engineering, University of Manchester, Manchester M13 9PL, UK

**Keywords:** EEV-based ejector, ejector-based multi-evaporator refrigeration system, refrigerator–freezer, entrainment ratio, coefficient of performance

## Abstract

This study presents an experimental rig of a multi-evaporator refrigeration system, in which the pressure difference between two evaporators can be maintained by using both the pressure-regulating valve (PRV) and electronic expansion valve (EEV)-based ejector. The proposed EEV-based ejector that is used to partially recover the throttling losses of the PRV consists of an EEV and the main body of an ejector. The established experimental system can work in both PRV-based mode and ejector-based mode by switching the valves. Via experimental means, the performances of both modes were evaluated by varying the cooling loads. Moreover, the effects of the spindle-blocking area percentage of the EEV-based ejector and the condensing temperature on the system performance were identified. The results showed that: (1) the system performance of the ejector-based mode was 3.6% higher than the PRV-based mode; (2) both entrainment ratio and coefficient of performance dropped along with the increase in ejector spindle-blocking area percentage; (3) compared to ejector spindle-blocking area percentage, the condensing temperature had a more evident influence on the system performance.

## 1. Introduction

Energy and the environment are crucial issues for all walks of life across the world [1,2]. According to the data from the National Bureau of Statistics of China, the production of household refrigerators in 2019 was 79.0 million units, with a rise of 6.3% year-on-year [3]. The household refrigerator occupies the main portion of household energy consumption [4]. More energy-efficient refrigerators are thus required in the market [5].

An ejector is a simple component that can mix two different pressure fluids and elevate the pressure of the mixed fluid without using mechanical energy [6]. In 1910, Leblanc designed an ejector-based refrigeration system [7]. Gay [8] presented a two-phase ejector instead of the traditional expansive valve to increase the system performance by reducing the throttling loss. Sokolov et al. [9] utilized the industrial waste heat as the heat source of the ejector-based refrigeration system, and presented an enhanced ejector-based refrigeration cycle. The new cycle had a performance boost in comparison with the common ejector-based refrigeration cycle. Croquer et al. [10] evaluated the ejector performance through thermodynamic modelling. Zhao et al. [11] proposed a novel system with a booster to elevate the secondary flow pressure and to further improve the system performance by using theoretical analysis. Hao et al. [12] presented a novel hybrid auto-cascade refrigeration system equipped with a heat-driven ejector cycle, and the system obtained an energy consumption reduction of 50% over the conventional system. Tan et al. [13] analyzed such a hybrid system performance with mixed refrigerants R32 and R236fa, and confirmed the system performance improvement.

Studies on ejector-based multi-evaporator refrigeration system (EMERS) for domestic refrigerators have been carried out by many investigators. Liu et al. [14] presented and compared three layouts for a hybrid compression-injection refrigeration system, and claimed that the compression-injection cross-regenerative hybrid refrigeration cycle had a 7.75% energy consumption reduction. Elakdhar et al. [15] performed studies on such systems, and their results showed that the ejection cycle could reach a higher COP and different refrigerants had different effects on the system performance. Wang et al. [16] presented a novel ejector-enhanced refrigeration system in a domestic refrigerator–freezer. The results showed that the system had a 5.45% energy consumption reduction compared to conventional systems. Zhou et al. [17] proposed a dual-nozzle ejector, which was used in the ejector-based refrigeration cycle for refrigerator–freezers. Compared to traditional ejectors, the novel dual-nozzle ejector can efficiently reduce the throttling losses. The COP improvement of the new cycle is 22.9%–50.8% greater than that of the conventional cycle.

However, refrigerators generally run in variable operating conditions, and thus the fixed-geometry ejector adopted in most ejector-based multi-evaporator refrigeration systems cannot work well. In order to solve this issue, Sun [18] analyzed the effect of variable ejector geometries for a 5 kW steam-jet refrigerator. The results showed that the variable-geometry ejectors could achieve better performance, and a guideline was presented to design such a system. Yan et al. [19] employed an air-cooled ejector cooling system to test the influences of six key geometry parameters of the ejector, and to seek the best design geometries through simulations. The results indicated that area ratio (AR) and nozzle exit position (NXP) were the most sensitive geometries. They also established an ejector cooling system with R134A to test the influence of the geometric parameters such as NXP, the length of constant-area section and diverging angle of primary nozzle [20]. Lin et al. [21] investigated the optimum geometry parameters of an adjustable ejector in a multi-evaporator refrigeration system, and the results indicated that the system obtained better compression energy efficiency with optimized geometry parameters. Li et al. [22] investigated the variable area ratio (AR, which is defined as the ratio of mixing chamber area to the primary nozzle throat area) ejector of a multi-evaporator refrigeration system. Key performance indicators, such as entrainment ratio, pressure recovery ratio, cooling capacity, and power consumption were identified by varying ejector area ratios. The results showed that the higher secondary pressure or lower primary pressure can achieve higher entrainment ratio. Moreover, the performance of the system was improved by up to 12% with the variable area ratio ejector compared to the conventional ejector-based refrigeration system. Hou et al. [23] investigated the performance of an adjustable ejector in a refrigerator–freezer refrigeration system through CFD simulations. The adjustable ejector, which was used to meet the requirements of variable cooling loads, could achieve high entrainment ratio and adapt higher back pressure. Thongtip et al. [24] studied the geometrical impact of the primary nozzle on the ejector performance in an ejector-based refrigerator by experimental method. The results showed that a bigger nozzle throat at lower generator operating temperature was preferable. Yongseok et al. [25] presented a modified two-phase ejector for an R600a household refrigeration cycle. The effect of the various ejector geometries on the performance under various operating conditions was investigated, and the performance of the cycle was also measured and analyzed. Zhang et al. [26] presented a tiny EEV-based ejector used in domestic refrigerator–freezers. The novel EEV-based ejector validated by the experimental results was highly adaptable to the variable cooling loads of the refrigerators, and the optimum geometry of the EEV-based ejector was obtained through CFD simulations. How does the ejector improve performance compared to the conventional system? The EEV-based ejector can regulate the mass flow to meet the variable cooling loads of the system. To be specific, the flow rate of the primary and secondary inlets of the ejector can be adjusted; thus, the cooling capacity of different evaporators can be altered. A schematic diagram and P-h chart for a two-evaporator refrigeration system with EEV-based ejector are shown as in Figure 1. By using an EEV-based ejector, the throttling loss of PRV can be partially recovered. The compressor power input can be reduced, and therefore the system performance can be improved. The working process of the system is described as below:The refrigerant enters the compressor at low pressure at state (1), and is compressed to the high-pressure point at state (2).The high-pressure refrigerant enters the condenser and is condensed to states (3) and (4), and then flows through expansion valves at states (5) and (6).Then, two-phase refrigerants enter two evaporators, and the refrigerants become superheated at states (7) and (8).Superheated refrigerants enter the EEV-based ejector and mix with each other.The mixed flow leaves the EEV-based ejector with a pressure lift and obtains the state (1), and the cycle is completed.

If the system is equipped with PRV only, it can be seen that the partial pressure lift cannot be obtained; hence, the throttling loss of PRV cannot be partially recovered.

In the previous studies, however, the experimental tests did not focus on the performance of the whole system.

To bridge the gap, therefore, this work aimed to carry out experimental study on the system performance of a multi-evaporator refrigeration system equipped with EEV-based ejector. The details of the study were as follows:First, a hybrid PRV- and ejector-based vapor-compression refrigeration system was established, in which PRV-based vapor-compression mode and ejector-based vapor-compression mode could be operated.Second, the performances of PRV- and ejector-based modes were identified through the experiments.Third, both the ejector and system performances were identified by varying the ejector spindle-blocking area percentage.Last, the system performance was verified by varying the condensing temperature.

## 2. Experimental Setup

The schematic of the proposed system is shown in Figure 2. The working process of the system is described as below:

The superheated and high-pressure vapor refrigerant compressed by the compressor flows through the oil separator and then is condensed in the condenser. The subcooled liquid refrigerant released from the condenser is dried and divided into two streams, which are expanded by EEV_1_ and EEV_2,_ and then evaporated in evaporator 1 and evaporator 2. Then, the two streams of superheated fluid from the evaporators go into the EEV-based ejector as the primary flow and secondary flow. The mixing process occurs in the EEV-based ejector, and the outlet flow enters in the accumulator. The low-pressure vapor is sucked into the compressor and the cycle repeats. The refrigerant used in this system is R600a for considering environmental protection.

The system can work in two modes, namely, PRV-based vapor-compression refrigeration mode and ejector-based vapor-compression refrigeration mode. Four valves are used to change the working mode. By switching on V_7_ and V_8_ and switching off V_4_ and V_5_, the system works in the PRV-based vapor-compression refrigeration mode; while by opening V_4_ and V_5_ and closing V_7_ and V_8_, the system can operate in the ejector-based vapor-compression refrigeration mode.

The sensors in the experimental platform and their relative errors are as follows [26]:PT1000 platinum resistance temperature sensors with error of ±0.3 °C;pressure transducers with full scale error of 0.5%;two oval wheal flow meters with error of ±0.5%.

Figure 3 shows the experimental platform of the multi-evaporator refrigeration system. Key components of this system, such as an EEV-based ejector, a closed compressor with power input 1.1 kW, an air-cooled condenser with maximum heat transfer 4.0 kW, two electric evaporators, several pressure transducers, a flow meter, and a temperature sensor, are illustrated in Figure 4. The evaporating temperature of evaporator 1 in chamber 1 used for cold storage is −5 °C, and the evaporating temperature of evaporator 2 in chamber 2 used for freezing is −30 °C.

In order to reduce the measuring errors, the reading values of every setting condition were recorded 5 times and their average values were selected. As for the uncertainty of the measurement, it mainly considered the precision limits of the experimental results caused by the accuracy of sensors. Uncertainty for temperature, pressure, power input and flow rate can be written as below:(1)$$U_{I}=\frac{P_{\left(I\right)}}{I}$$
where

*U_I_*: uncertainty for temperature, pressure, power input, and flow rate;

*P*_(*I*)_: uncertainty due to accuracy;

*I:* measured temperature, pressure, power input, and flow rate.

The uncertainty for ER is as follows:(2)$${\left(U_{ER}\right)}^{2}={\left(\frac{P_{\left(ER\right)}}{ER}\right)}^{2}={\left(\frac{P_{\left(M_{p}\right)}}{M_{p}}\right)}^{2}+{\left(\frac{P_{\left(M_{s}\right)}}{M_{s}}\right)}^{2}$$
where

$U_{ER}=\frac{P_{\left(ER\right)}}{ER}$: uncertainty for ER;

$\frac{P_{\left(M_{p}\right)}}{M_{p}}$: uncertainty for primary flow rate;

$\frac{P_{\left(M_{s}\right)}}{M_{s}}$: uncertainty for secondary flow rate.

The uncertainty for COP can be derived in the similar way to Equation (2).

The uncertainties of 9 parameters (T4, T5, P4, P5, *m_p_*, *m_s_*, *W_c_*, ER, COP) were obtained with values of ±5.99%, ±1.07%, ±4.63%, ±10.37%, ±1.63%, ±2.45%, ±0.91%, ±2.95%, and ±3.08%, respectively.

## 3. Geometrical Details of the EEV-Based Ejector

Figure 5 presents the external view of the EEV-based ejector that is welded by the majority of an EEV and the main part of an ejector without the primary nozzle. Figure 6 displays the cross-sectional view of the EEV-based ejector, and its geometrical details are presented in Ref. [26]. The EEV used in this study was produced by Sanhua company. The refrigerant mass flow rate flowing through the EEV-based ejector can be regulated by tuning the spindle of the EEV.

## 4. Results and Discussion

### 4.1. System Performance of Both Modes

The COP is the main indicator of the system performance, which is defined as below:(3)$$COP=\frac{Q_{c}}{W_{c}}$$
where $Q_{c}$ is the cooling capacity of the whole system, which consists of two parts: one is the cooling capacity of the refrigerating chamber, and the other is that of the freezing chamber. With thermal balance measurement, both of them can be measured by using two power meters, which are installed in the control cabinet. $W_{c}$ is the power input of the compressor, which is also measured by a power meter.

Once the system was operating steadily in the PRV-based refrigeration mode, 19 random experimental datasets with different cooling capacities were selected to evaluate the system performance, in which the cooling capacity and COP of the mode were recorded and calculated. In addition, the relevant data in the ejector-based refrigeration mode were recorded. These experimental data were taken to verify the COP comparison between the PRV-based refrigeration mode and the ejector-based refrigeration mode, as displayed in Figure 7. It can be observed that with variable cooling capacity, the COPs of the PRV-based refrigeration mode and the ejector refrigeration mode had obvious distinctions. The results show that the COP of both modes rose with the increase in the cooling capacity, and the COP of the ejector-based mode was 3.6% higher than the PRV-based vapor-compression refrigeration mode.

### 4.2. Effect of Spindle-Blocking Percentage on the Ejector Entrainment Ratio and System Performance

In the ejector-based refrigeration mode, the spindle of the EEV part in the EEV-based ejector is used to adjust the flow of ejector. In this section, both the ejector entrainment ratio (ER) and the COP of the system were taken as the effect indicators. The ER can be calculated by the primary mass flow and secondary mass flow of the ejector and is defined as below:(4)$$ER=\frac{m_{s}}{m_{p}}$$
where $m_{p}$ is the mass flow rate of the primary flow measured by the flowmeter 1, and $m_{s}$ is the mass flow rate of the secondary flow measured by the flowmeter 2. Seven different blocking percentages (ranging from 0 to 60% with steps of 10%) were set by tuning the spindle of the EEV. The variations of ER and COP with spindle-blocking percentage (A/A_n_, where A is the cross-sectional area of primary nozzle throat with spindle blocked; A_n_ is the initial cross-sectional area of primary nozzle throat) are shown in Figure 8 and Figure 9, respectively. It was revealed that both the ER and COP dropped with the rise in the spindle-blocking percentage. To be specific, when the spindle-blocking area varied from 0 to 60%, ER dropped from 1.25 to 0.6, and COP dropped from 1.49 to 1.03. The effect of the blocking percentage seemed more visible in the ER than COP.

### 4.3. Effect of Condensing Temperature on System Performance

The condensing temperature is one of the influencing factors in the refrigeration system. With the ejector-based refrigeration mode, in this section, the effect of the condensing temperature with different ejector spindle-blocking area percentages on the system performance was taken into consideration. The condensing temperature was controlled by using a heat pump in the test room. The COP was recorded by regulating the temperature of the condenser. The results are shown in Figure 10. For a given spindle-blocking area percentage, the system COP decreased as the condensing temperature increased. With a spindle-blocking area percentage of 20%, for example, the COP dropped from 1.43 to 1.21 when the condensing temperature varied from 35 °C to 40 °C. Moreover, for a given condensing temperature, the COP decreased along with the increase in the spindle-blocking area percentage, which was similar to the results in Section 4.2. According to the maximum relative deviation of COP to ejector spindle-blocking area percentage and condensing temperature, it can be inferred that the condensing temperature had a noticeable impact on the system COP as compared to the ejector spindle-blocking area percentage. In order to improve the system performance, lower ambient temperature is preferred.

## 5. Conclusions

In this paper, a multi-evaporator refrigeration system with EEV-based ejector was established to meet the variable working conditions of domestic refrigerator–freezers. The system can work in both PRV-based mode and ejector-based mode, and the performances of both modes were evaluated accordingly. Meanwhile, the influence on the system performance of the ejector spindle-blocking area percentage, as well as the condensing temperature, was identified. The primary findings observed are as follows:(1)With the increase in the cooling capacity, the COP of both modes increased. The ejector-based mode had a COP 3.6% higher than the PRV-based vapor-compression mode.(2)With the increase in the ejector spindle-blocking percentage, ER dropped by 52% and COP dropped by 30.8%. The effect of the blocking percentage on ER seemed more evident than that on COP.(3)The system COP decreased along with both the spindle-blocking area percentage and condensing temperature. In comparison, the system COP was more sensitive to the condensing temperature than the ejector spindle-blocking area percentage. Lower condensing temperature and spindle-blocking area percentage of the ejector offered better system performance.

## Figures and Tables

**Figure 1 entropy-24-01302-f001:**
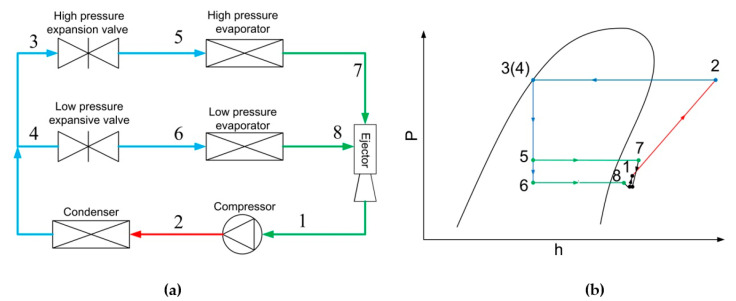
Schematic and P-h chart of a two-evaporator refrigeration system with EEV-based ejector: (**a**) Schematic; (**b**) P-h chart.

**Figure 2 entropy-24-01302-f002:**
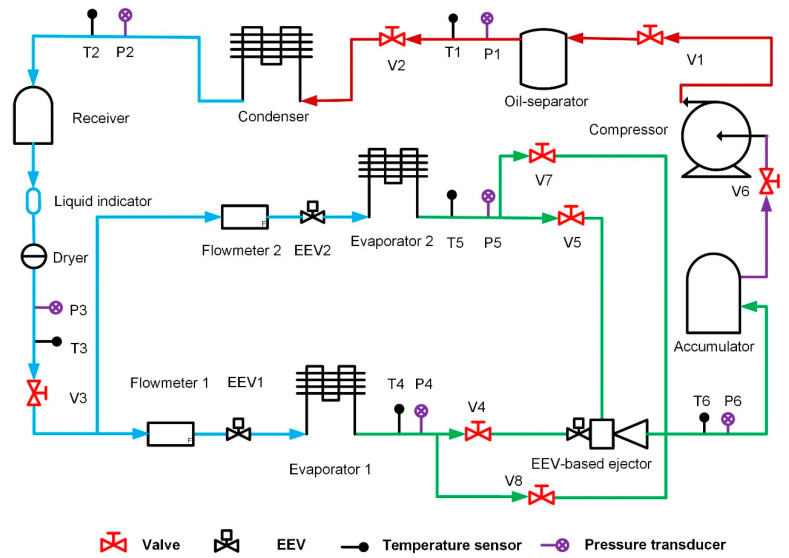
Schematic of the multi-evaporator refrigeration system with an EEV-based ejector.

**Figure 3 entropy-24-01302-f003:**
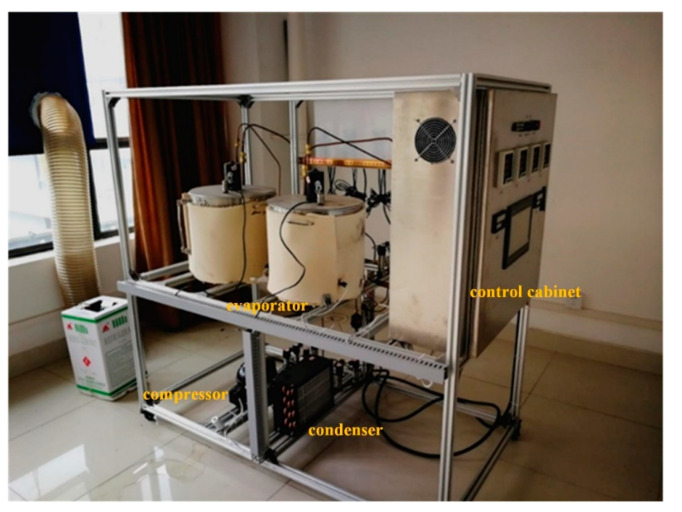
Experimental platform of the multi-evaporator refrigeration system.

**Figure 4 entropy-24-01302-f004:**
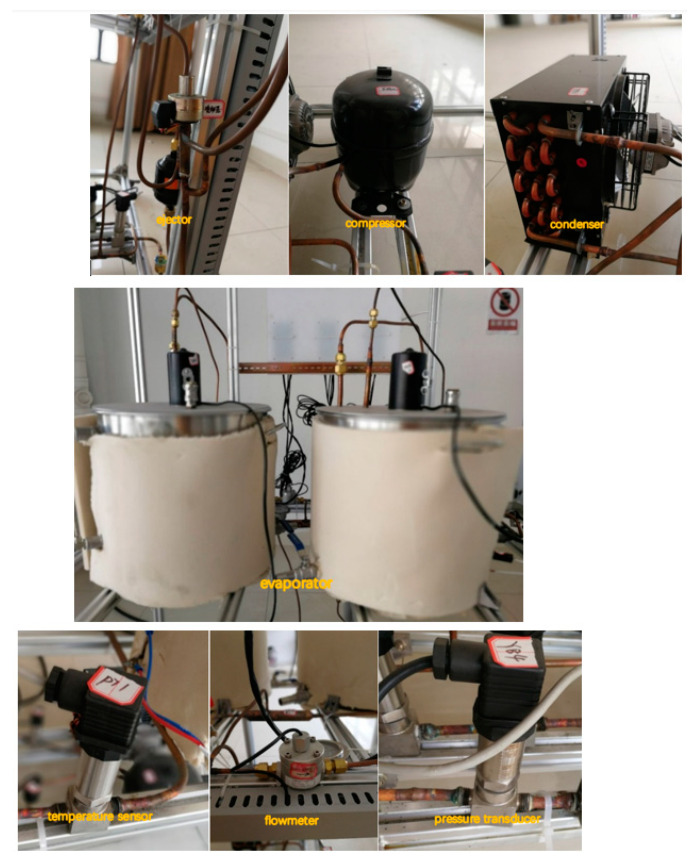
Components of the multi-evaporator refrigeration system.

**Figure 5 entropy-24-01302-f005:**
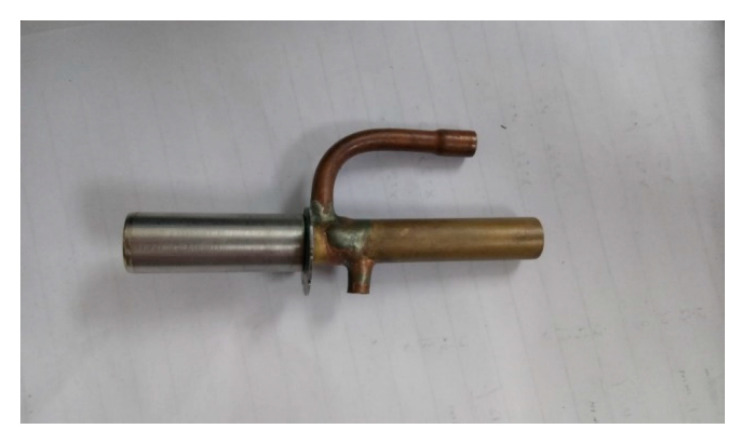
The external view of the welded EEV-based ejector.

**Figure 6 entropy-24-01302-f006:**
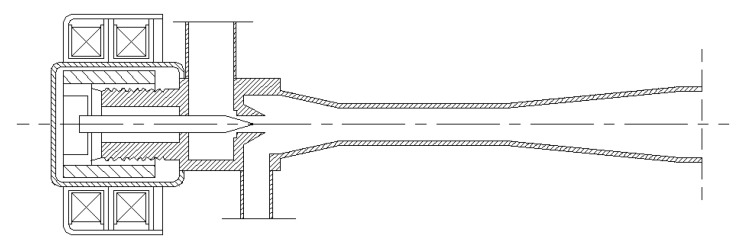
The cross-sectional diagram of the welded EEV-based ejector.

**Figure 7 entropy-24-01302-f007:**
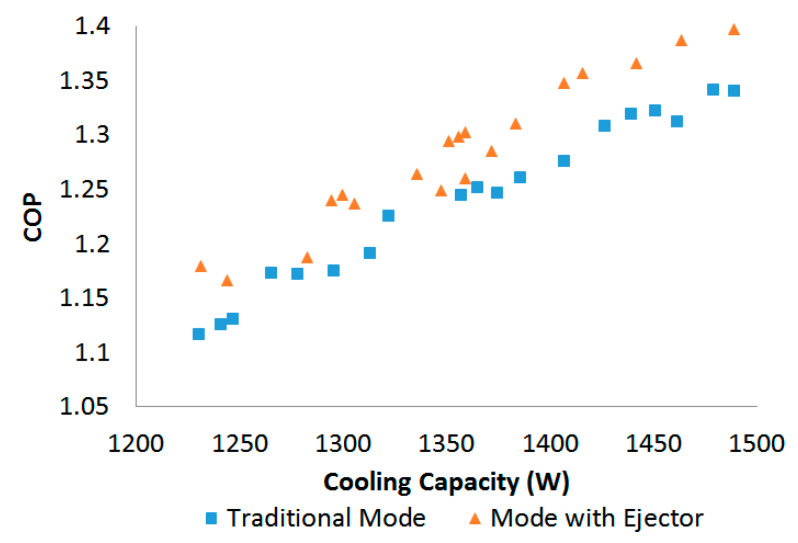
System performance of the PRV-based and ejector-based mode.

**Figure 8 entropy-24-01302-f008:**
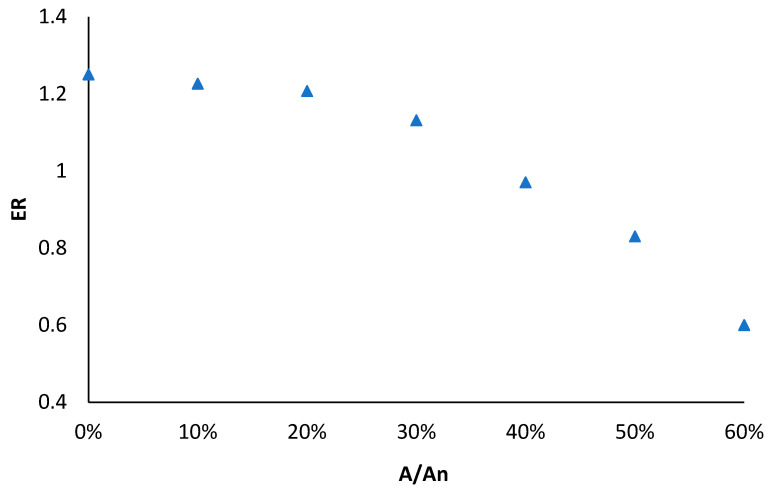
Variation of ejector entrainment ratio with spindle-blocking percentage (A/A_n_).

**Figure 9 entropy-24-01302-f009:**
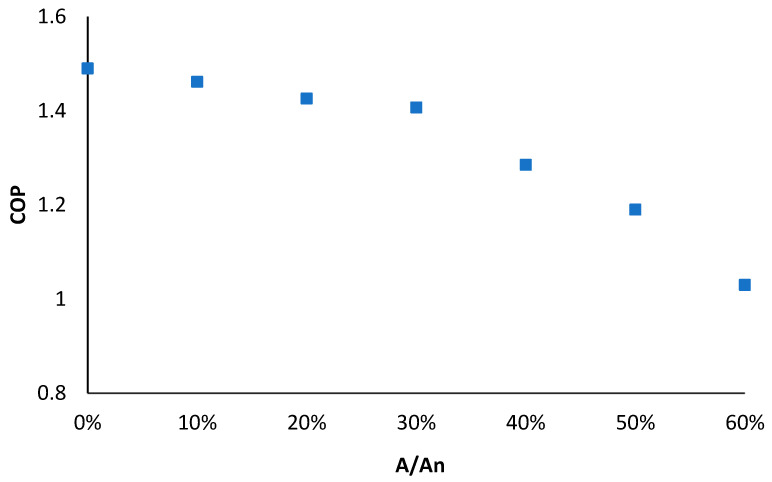
Variation of COP with spindle-blocking percentage (A/A_n_).

**Figure 10 entropy-24-01302-f010:**
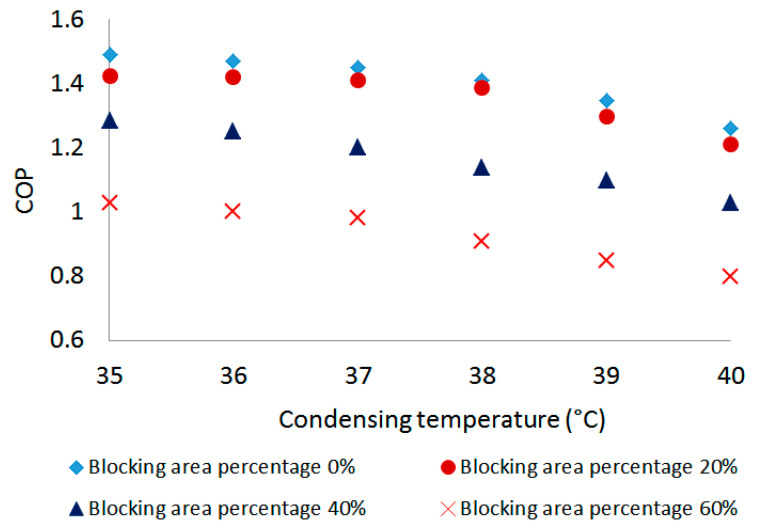
Variation of COP with condensing temperature.

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
