# Peer review of "Experimental Study on a Multi-Evaporator Refrigeration System Equipped with EEV-Based Ejector"

_entropy, 2022, doi:10.3390/e24091302_

Round 1
Reviewer 1 Report
The manuscript presents an experimental investigation of a multi-evaporator refrigeration system. Two-cycle configurations are presented: pressure regulating valve system and ejector system.
My comments are:
1. Line 49: the authors mentioned a novel improved system with a booster. Please, in few words describe the system and provide the improved performance.
2. Line 54: any performance related to the system analyzed by Tan?
3. Please define all acronyms AR, A, An…
4. Line 101: please cite the reference properly, ‘’ Zhang et al. [26]…’’
5. During the experiment, any special action, equipment...was used regarding the flammable side of the refrigerant R600a?
6. Please give the uncertainties on the mass flow rate, COP, ER
7. With PRV-based mode, the outlet of the two evaporators that are at different pressure levels meet before the accumulator. Have you noticed any issues at this point?
8. Regarding the flow meter location in the system, have you noticed any trouble due to possible phase change?
9. Line 159: Data is an average value of 5 records, but with which time interval?
10. How do you design the ejector body?
11. By using a straight section primary nozzle and not a convergent-divergent nozzle, do you think it will penalize the performance of the ejector?
12. Evaporator 1 feeds the primary of the ejector, hence P4/P5 (Pprimary/Psecondary) is probably very low, is it sufficient to activate the ejector?
13. Figure 7: change W by Qc
14. Figure 7: with what conditions at the condenser, evaporators 1 and 2 figure 7 was plotted?
15. Line 191: ‘’…the COP of the ejector-based mode are 3.6% higher than the PRV…’’, the 3.6% is an average of all the points measured or it is for a particular point?
16. Figure 7: The COP improvement of the ejector system needs a more detailed explanation with more results. For example, is it due to a reduction in compressor consumption or an improvement in evaporator capacity? Does the ejector increase the pressure at the suction of the compressor?
17. Define ejector spindle blocking area percentage
18. Figures 8-9: please provide the experimental conditions (condenser, evap1, evap2)
19. To have a better idea of the ejector performance, please add to figure 8 the compression ratio (P6/P5).
20. Figure 10: Do the conditions at the evaporators (Pevap, capacity) change with the condensing temperature?
21. Figure 10: For different condensing temperature, the ejector spindle blocking area of 0% gives the maximum COP. Does this mean that the spindle is not necessary in this case?
Reviewer 2 Report
The paper proposed the experimental work on the multi evaporator based refrigeration system in which the ejector-based EEV is used to enhance the whole system performance. The topic is interesting and worthy of investigation. However, more scientific contribution is required and many points still needs clarifications and substantial improvement so that the paper is reconsidered for publication. The reviewer provides comments as follows.
Comments on Introduction
- the authors must clearly mention the reason why the proposed EEV based ejector is worthy of investigation. This is because the authors does not demonstrate how the ejector can improve the performance when comparing the conventional system.
- the author must mention clearly how the proposed EEV based ejector can give advantage over the PRV based (show the process of the two system on T-s diagram or P-h diagram). This is to provide the background based on thermodynamic cycle and to convince the reader.
- the author should clearly mention why the EEV based is necessary for the multi evaporator based refrigeration system.
- for the literature review, the authors provide several literature reviews, however they does not reflect the research gap in this research field. Hence, the authors should provide comments and brief discussion of such the published papers so that it will reflect the new information of the submitted manuscript.
- since the EEV-based ejector does not new technology for cycle improvement (it was studied previously by many researchers or even the authors' previous work), the authors must clearly explain why this technology is still considered for the author’s research.
Comments on experimental setup part,
- More details of constructing the test rig and specifications of the major components must be added for being a reference case.
- the uncertainty analysis of the relevant parameters (obtained from measurement) must be implemented. This is to provide the reasonable accurate results.
- since the EEV-based ejector is the highlight of the submitted manuscript, however, the authors did not provide the criteria to design and to modify it. The reviewer think the spindle of EEV is commercially available (if not, please provide details).
- the spindle nozzle part as proposed by the authors (see from Fig.6) seems to be working similar to the converging nozzle which is mostly used for two-phase or liquid nozzle. However, the working principle of the proposed system is based on the vapor ejector in which the supersonic nozzle (converging-diverging nozzle) is more suitable. Thus, the authors must clearly explain why the authors designs the spindle nozzle to operate based on the converging nozzle which provide a poorer performance.
- the percent blockage can alternatively indicate the flow area of the nozzle throat to the nozzle exit diameter. This will provide more valuable for being a reference case. The authors must consider this parameter for providing more scientific contribution.
- the authors did not mention why the ejector throat diameter of 3.9 mm to be used (please explain to be consistent with the cooling capacity)
- evaporating temperature of the two evaporator did not provide clearly. This indicate that the discussion of the results doesn’t provide accurately.
- please provide the detailed design of the evaporators and provide technical information for controlling the evaporating temperature during the experiment.
Comments on discussion part,
- since the authors placed the flow meters at the two inlet ports of the ejector, the mass flow rate of the primary and secondary fluid must be reported especially for the mass entrainment ratio against percent blockage of the spindle.
- the authors only report the COP without the discussion on the developed EEV-based ejector performance, and, thus, the ejector performance indicator (mass entrainment ratio, pressure lift ratio, ejector efficiency against the variation of the evaporating temperature (for two evaporators) must be added for providing more scientific contribution.
- Since the authors placed the pressure transducer or pressure gauge (P6) at the outlet of the ejector, the authors must report the entrainment ratio against the ejector discharge pressure or pressure lift ratio. This is to provide reference case and provide insight into the ejector application based on two evaporators vapor compression system.
- the authors must consider to add the results of the exergy destruction or entropy generation for providing more contribution and to be consistent with the journal scope and special issue’s scope. This will help to convince more audiences.
Reviewer 3 Report
The submitted article presents original applied research results, thus its publication can be recommended.
Some minor corrections are suggested in the attached file.

Round 2
Reviewer 2 Report
Significant improvement of the manuscript have been made by the authors, and, therefore, the current form can be accepted for publication.